# The Impact of Short-Chain Fatty Acids on Neonatal Regulatory T Cells

**DOI:** 10.3390/nu14183670

**Published:** 2022-09-06

**Authors:** Jessica Chun, Gergely Toldi

**Affiliations:** 1School of Biological Sciences, University of Auckland, Auckland 1010, New Zealand; 2Liggins Institute, University of Auckland, Auckland 1023, New Zealand

**Keywords:** SCFA, butyrate, acetate, propionate, neonate, newborn, regulatory T cells

## Abstract

Over the first weeks of life, the neonatal gastrointestinal tract is rapidly colonised by a diverse range of microbial species that come to form the ‘gut microbiota’. Microbial colonisation of the neonatal gut is a well-established regulator of several physiological processes that contribute to immunological protection in postnatal life, including the development of the intestinal mucosa and adaptive immunity. However, the specific microbiota-derived signals that mediate these processes have not yet been fully characterised. Accumulating evidence suggests short-chain fatty acids (SCFAs), end-products of intestinal bacterial metabolism, as one of the key mediators of immune development in early life. Critical to neonatal health is the development of regulatory T (Treg) cells that promote and maintain immunological tolerance against self and innocuous antigens. Several studies have shown that SCFAs can induce the differentiation and expansion of Tregs but also mediate pathological effects in abnormal amounts. However, the exact mechanisms through which SCFAs regulate Treg development and pathologies in early life remain poorly defined. In this review, we summarise the current knowledge surrounding SCFAs and their potential impact on the neonatal immune system with a particular focus on Tregs, and the possible mechanisms through which SCFAs achieve their immune modulatory effect.

## 1. Introduction

At birth, the immunologically naïve neonate undergoes a drastic transition from the sterile environment of the maternal womb to the microbially rich extrauterine environment. Consequently, newborns are highly vulnerable to infections, which account for 40% of the 3 million neonatal deaths worldwide each year [1]. In addition, the first few weeks of life represent a period in which rapid and significant microbial colonisation of the neonatal gastrointestinal tract occurs [2]. As a result, the neonatal immune system is bombarded with a diverse array of novel antigens, which induce dynamic adaptive changes in immune function to accommodate the acquisition of symbiotic microbes while retaining the capacity to protect against infectious challenge. Microbial colonisation of the neonatal gut is a well-established regulator of several physiological processes that contribute to immunological protection in postnatal life, including the development of the intestinal mucosa and adaptive immunity [2,3]. However, the specific microbiota-derived signals that mediate these processes have not yet been fully characterised. Accumulating evidence suggests short-chain fatty acids (SCFAs), end-products of intestinal bacterial metabolism, as one of the key mediators of neonatal immune development. 

Of particular importance during immune development in early life is the establishment of tolerance against self and innocuous antigens derived from exogenous sources such as nutrition, commensal bacteria and the mother. The suppression of immune activation against these antigens is largely mediated by regulatory T (Treg) cells, which represent a subset of CD4+ T cells that are characterised by their immune suppressive effects [4]. Various studies have shown that SCFAs can induce the differentiation and expansion of Tregs, but the exact mechanisms through which SCFAs regulate Treg development and the extent of their effects in the neonate remain poorly defined. Moreover, mounting evidence indicates that SCFAs may also yield pathological effects in neonates when present in abnormal amounts. 

In this review paper, we summarise the current knowledge surrounding SCFAs and their potential impact on the neonatal immune system with a particular focus on regulatory T cells and the possible mechanisms through which SCFAs achieve their immune modulatory effect. 

## 2. T Cells in the Neonatal Period

T cells are specialised lymphocytes that play a pivotal role in the adaptive immune response and are marked by their surface expression of a T cell receptor (TCR). Conventional T cells are classically divided into one of two major subtypes based on the identity of their TCR co-receptor: CD4+ T cells or T helper (Th) cells play a key role in orchestrating adaptive immune responses via the production of effector cytokines; CD8+ T cells, also known as cytotoxic T lymphocytes (CTLs), are critical mediators in the elimination of virally infected or tumour cells, which they achieve through the release of cytotoxic granules that induce apoptotic or lytic death. Together, these T cell subpopulations coordinate immune responses that protect the host from infections, cancer and autoimmunity. 

T cell development begins during early embryogenesis. T cell precursors are initially born from the fetal liver, followed by haematopoietic stem cells (HSCs) in the bone marrow [5]. After 7 weeks of gestation, CD34+ T cell precursor cells migrate to the thymus, where most differentiate to express a uniquely rearranged, antigen-specific αβ TCR and a CD4 or CD8 co-receptor [5]. Within the thymus, these TCR+ thymocytes undergo a process called central tolerance, in which T cells capable of recognising self-major histocompatibility complex (MHC) molecules are selected for survival (positive selection), and those reactive to self-antigens presented by thymic cells are selectively, albeit imperfectly, eliminated (negative selection) [6]. T cells first exit the thymus in a phenotypically and functionally immature state, referred to as recent thymic emigrants (RTEs), at around 12–14 weeks of gestation [7,8]. These RTEs then enter an approximately 3-week period of post-thymic maturation to give rise to a more mature naïve (MN) T cell compartment [7]. Although diversification of the TCR repertoire is completed by ~26 weeks of gestation, the early-life T cell compartment continues to evolve even after parturition and does not acquire its adult phenotype until at least 2 years of age [9,10]. This is in contrast to other immune cell types (e.g., B cells, natural killer cells, dendritic cells), which come to resemble their adult counterparts by the first 3 months of life [11]. 

## 3. Neonatal CD4+ T Cells

Work over the past three decades has revealed significant heterogeneity within the CD4+ T cell compartment in terms of their immunophenotype [12]. Depending on the cytokine milieu at the time of TCR activation, CD4+ T cells can differentiate into one of several different subtypes, each of which expresses a characteristic lineage-specifying transcription factor (TF) and produce a unique set of cytokines that mediate distinct functions in immunity. The five most well-established subtypes include Th1, Th2, Th17, T follicular helper (Tfh) and Treg cells (Figure 1). Th1 cells express the master TF T-bet and produce interferon (IFN)-γ and interleukin (IL)-2, which promote cell-mediated immunity to intracellular pathogens such as viruses and mycobacteria and are implicated in autoimmunity. Th2 cells, defined by their expression of GATA3, produce IL-4, IL-5 and IL-13 to promote humoral immunity to large extracellular pathogens such as helminths and are associated with asthma and allergies. Th17 cells, which express RORγt, produce IL-17 and IL-22 to mediate strong pro-inflammatory epithelial responses to select fungi and extracellular bacteria but are also known to drive extensive inflammation and autoimmunity. Bcl6-expressing follicular T helper (Tfh) cells produce IL-21, which supports the production of antibodies by B cells. Lastly, Treg cells express Foxp3 and produce transforming growth factor (TGF)-β and IL-10, which, unlike the cytokines produced by Th1/Th2/Th17/Tfh cells, play a critical role in regulating the immune response to maintain self-tolerance and prevent immunopathology.

Seminal studies by Medawar and colleagues demonstrated that exposure to allogeneic cells at birth induces mice to become tolerant to transplants expressing the same alloantigens as adults [13]. This phenomenon, which has since been termed ‘neonatal tolerance’, was initially associated with intrathymic clonal deletion of allospecific CD8+ CTLs [14,15]. However, later studies revealed that a biased CD4+ Th cell response may also be involved. Indeed, anti-donor CD4+ Th cells in neonatal tolerant mice preferentially produce Th2 cytokines (IL-4) at the expense of inflammatory Th1 cytokines (IL-2 and IFN-y) [16,17,18]. This was found to be due to IL-4-mediated apoptosis of alloreactive Th1 cells [19]. Debock et al. further demonstrated that the inhibitory effect of IL-4 in neonatal tolerance extends to the generation of inflammatory Th17 responses [20]. IL-4 deprivation of neonatal mice immunised with allogeneic cells led to impaired development of allospecific Th2 responses and upregulation of mRNA encoding RORγt and Th17-type cytokines, which were associated with increased differentiation of donor-specific Th17 cells and graft rejection. These observations indicate that the neonatal T cell compartment is skewed toward the Th2 phenotype, which impairs the development of inflammatory Th1/Th17 responses. Moreover, they demonstrate that neonatal T cells are capable of eliciting effective, albeit biased, immune responses, challenging the traditional dogma that they are immunodeficient. Further support for neonatal immunocompetence comes from murine studies demonstrating the ability of newborns to mount adult-like protective Th1/Th17 responses under certain conditions [21,22,23,24]. A Th2-biased immune system also better explains why neonates are more susceptible to both infection and allergic diseases and is more sensible in terms of teleology, as excessive Th1/Th17 inflammation would be deleterious to the antigen-bombarded newborn [25]. 

## 4. Neonatal Regulatory T Cells

As immune deviation to Th2 suppresses Th1/Th17 responses in early life, adaptive immune responses during the neonatal period are typically considered to be anti-inflammatory. Critical to the maintenance of this anti-inflammatory status are Treg cells, which are highly prevalent in the cord blood CD4+ T cell compartment (~12%) and neonatal lymphoid tissues (~8%) [26]. In utero, Tregs play an essential role in preventing fetal inflammation and rejection of semi-allogeneic maternal tissue and/or cells since they dampen pro-inflammatory T cell activity and promote tolerance to maternal antigens [27]. Recent work by Wood et al. also demonstrated that the development of immune tolerance to breastmilk-derived non-inherited maternal antigens (NIMA) in early life was mediated by neonatal Tregs [28]. CD4+ CD25+ Treg cells also negatively regulate cytotoxic activity by alloreactive CD8+ CTLs in neonatal tolerant mice [29]. Moreover, Treg and Th17 cell differentiation are reciprocally regulated [30]. While Foxp3 and RORγt both require TGF-β for upregulation, in the absence of pro-inflammatory cytokines (e.g., IL-6), the Treg master TF dominates and Th17 differentiation is suppressed. This may represent a potential mechanism by which Th17 activity is downregulated in early life. 

Accumulating evidence suggests that T cell immunity in early life exhibits a strong bias towards Treg development. For instance, fetal naïve CD4+ T cells proliferate more rapidly and exhibit a greater propensity for Treg differentiation following TCR stimulation compared to their adult counterparts [31]. In line with this, fetal lymphoid tissues are enriched in tolerogenic cytokines that favour Treg differentiation over pro-inflammatory T cell activity, including members of the TGF-β family, which are known to induce Foxp3 upregulation [32]. The TF Helios, which is not expressed in adult naïve T cells, was found to play an important role in contributing to this predisposition, as Helios-deficient fetal naïve CD4+ T cells were unable to differentiate into Treg cells [33]. A similar propensity is evident in neonates. Wang et al. demonstrated that CD4+ thymocytes and T cells in neonatal mice display a ‘default’ tendency to differentiate into Foxp3+ Treg cells irrespective of the TCR stimulus and without the need for exogenous addition of TGF-β [34]. In humans, naïve T cells in cord blood were shown to be more prone to differentiating into functional Foxp3+PD-1+ Treg cells in an antigen-presenting cell (APC)-dependent manner compared to those in adult peripheral blood [35]. These findings indicate that Treg cells play an important role in regulating immune reactivity early in life. However, despite their protective effects, Treg cells have also been implicated in increased susceptibility to infections and dampened vaccine efficacy in neonates. Hence, further research into neonatal Treg cells is crucial to gain a deeper understanding of their function and role in health and disease [8]. 

## 5. Factors Influencing the Neonatal Regulatory T Cell Compartment 

Over the first weeks of life, neonates are confronted with an enormous array of novel antigens that pose a significant homeostatic challenge to their developing immune system. The primary sources of these immunostimulants are the microbes that rapidly colonise the neonatal gut at parturition through exposure to the mother’s vaginal and skin microbiome or hospital-associated surfaces, which subsequently undergo dynamic alterations in their composition in response to factors such as diet (e.g., breastfeeding) or antibiotic administration [36]. A growing body of research suggests that these early-life microbial exposures can permanently program the neonatal immune system and modify susceptibility to various diseases in adult life [37,38]. For instance, the dynamic changes that occur in the intestinal microbiota during the neonatal period are known to influence the maturation and activity of gut immune cells such that they promote host-microbiome symbiosis [39]. In line with this, perinatal colonisation of the gut microbiome has been well-established as a dominant regulator of neonatal immune development, including the maturation of immunosuppressive Treg cells [40,41,42]. Moreover, dysbiosis in the neonatal period is associated with both acute and chronic immune dysregulation, leading to conditions such as necrotising enterocolitis or various allergic, inflammatory and metabolic diseases in later life [43]. 

Commensal bacteria of the gut microbiota interact with the mucosal and systemic immune systems largely via compounds derived from microbial metabolism, which act as signals in molecular pathways that mediate cellular responses such as immune development [44]. The molecular recognition of these microbial metabolites by the neonatal immune system also likely regulates Treg maturation and Treg-mediated immune tolerance in early life. However, the specific microbiota-derived signals that mediate these processes and the mechanisms through which they achieve these have not yet been fully described. 

In previous studies, short-chain fatty acids (SCFAs) have been implicated in promoting Treg differentiation and proliferation [41,45]. Short-chain fatty acids (SCFAs) are carboxylic acids with aliphatic tails of 1–5 carbon atoms, of which the most abundant are acetate (C2), propionate (C3) and butyrate (C4) [46]. SCFAs are produced through the fermentation of undigested polysaccharides by certain bacterial members of the gut microbiota—in particular, the *Actinobacteria*, *Bacteroidetes*, *Firmicutes*, *Spirochaetes* and *Verrucomicrobia* phyla [46,47]. In neonates, human milk oligosaccharides (HMOs) are a significant source of SCFA production, as discussed later in this review.

Most of the SCFAs produced in the gut (~95%) are rapidly absorbed through the intestinal mucosa or used as an energy source by colonocytes [48]. At the local level, SCFAs are crucial for maintaining intestinal epithelium physiology through the regulation of cellular turnover and barrier functions [47]. Given their ability to be transported into the circulation and distributed via the blood, there is accumulating evidence that SCFAs may also have wider systemic effects in organs distal to the intestine [49]. In particular, SCFAs have been shown to play an important role in regulating the activation, recruitment and differentiation of immune cells, including Tregs, to promote anti-inflammatory immune responses [49]. For instance, SCFAs can have direct effects on T cells, leading to reduced proliferation and increased differentiation toward a regulatory phenotype [50]. Moreover, butyrate can condition murine and human dendritic cells (DCs) to promote the differentiation and expansion of Tregs [47,50]. Recent studies have also demonstrated that SCFAs have a potential role in directly inducing Tregs in the gut [41,51,52].

SCFAs exert their anti-inflammatory and immunomodulatory effects via two main mechanisms. The first involves signalling through specific G-protein coupled receptors (GPCRs) on the target cell surface, the most well-characterised SCFA-sensing GPCRs being GPR41, GPR43 and GPR109A [53]. 

The second mechanism involves the inhibition of histone deacetylases (HDACs) to regulate gene expression. HDAC inhibitor activity is particularly characteristic of butyrate and propionate [46]. HDACs, together with histone acetylases (HATs), control the acetylation of lysine residues within histones, which plays a key role in the epigenetic regulation of gene expression by facilitating the access of transcription factors to promoter regions. HDACs remove acetyl groups from histones, thus promoting more repressive chromatin—as such, inhibition of HDAC activity can enhance gene transcription by increasing histone acetylation. While this can result in a wide range of downstream effects, most studies on immune cells have linked SCFA-mediated inhibition of HDACs to the downregulation of inflammatory responses [47,54,55,56].

## 6. SCFA Levels in the Neonate

Current understanding of the effects of SCFAs on Tregs during the neonatal period is limited, as most studies concerning the immunomodulatory effects of SCFAs have been largely conducted in adult animals.

Adults derive the substrates for SCFA production from the ingestion of dietary fibre, but for newborns, consumption of solid foods containing dietary fibre does not begin until the weaning period. It is therefore likely that neonates derive SCFAs or the substrates for SCFA production from the maternal breastmilk or formula feed. Maternal breastmilk is the optimal nutritional supply for the newborn infant [57].

Recent work by Stinson et al. demonstrated that human breastmilk contains detectable levels of the SCFAs acetate, butyrate and formate at 1 month postpartum [58], which is in line with previous studies of the human milk metabolome [59,60]. All of these SCFAs were also detected in the breastmilk samples from a single woman using nuclear magnetic resonance (NMR) as early as 24 days postpartum [61], suggestive of the potential role of breastmilk SCFAs in the neonatal period. However, further studies with a larger cohort of women are required to characterise the SCFA profile of human breastmilk in the early lactation stage. Breastmilk SCFAs are likely produced by the maternal gut microbiota and distributed to the mammary gland via the circulation. They may also be produced by the broad range of bacteria that are resident in the human breastmilk, although evidence for this possibility is currently lacking [58].

Human breastmilk also contains a significant amount of complex non-digestible carbohydrates, which are collectively called human milk oligosaccharides (HMOs). HMOs serve as preferred substrates for certain gut microbiota in SCFA production, including certain species of *Bifidobacterium* [62]. *Bifidobacterium*, being the predominant bacterial genera in the HMO-enriched guts of breastfed neonates, have been implicated in directing immune system development in early life [63]. In line with this, infants colonised with *Bifidobacteria* are known to produce high levels of SCFAs [64], which is unsurprising as the primary products of *Bifidobacterium* fermentation are acetate and lactate [65].

## 7. The Clinical Role of SCFAs in the Neonate

The role of SCFAs has been well documented in immune-mediated disorders in adults, most extensively in asthma. For instance, evidence from mouse models indicates an association between increased maternal dietary microbiota-accessible carbohydrates, SCFA exposure during pregnancy, and reduced offspring asthma mediated by the induction of Tregs in the lung [66]. Human breastmilk samples from atopic mothers had significantly lower concentrations of acetate and butyrate than those of non-atopic mothers. This reduced exposure to human milk SCFAs in early life may program atopy or overweight risk in breastfed infants [58,59].

In contrast, the only neonatal complication where the contribution of SCFAs was investigated is necrotising enterocolitis (NEC), an inflammatory condition of the bowels characterised by decreased epithelial barrier function, translocation of gut bacteria causing sepsis, and perforation of the intestine. He et al. examined the effects of human-to-mouse fecal microbiota transplants (FMT) on intestinal histological injury in mice receiving the microbiome isolated from fecal samples of patients with NEC and control infants matched by gestational age, birth weight, date of birth, mode of delivery and feeding patterns. FMT in germ-free mice with samples from NEC patients achieved higher histological injury scores when compared to mice that received FMT with control samples. The prevalence of Treg cells was reduced in both NEC patients and mice modelling NEC following FMT. NEC patients had increased *Proteobacteria* and decreased SCFA-producing *Firmicutes* and *Bacteroidetes* compared to fecal control samples, and the level of butyrate in the NEC group was lower than the control group. Alterations in microbiota and butyrate levels were maintained in mice following FMT [67].

Roy et al. developed a piglet model which replicates neonatal NEC with the aim of characterising the importance of bacterial fermentation of formula and SCFAs in its pathogenesis. SCFA levels were increased in *Escherichia coli*-fermented formula-treated porcine bowels, which demonstrated inflammation, coagulative necrosis and pneumatosis resembling human NEC. The authors concluded that while *E. coli* treatment alone can initiate intestinal inflammation, injury and apoptosis, bacterial fermentation of formula by *E. coli* generates SCFAs, which contribute to the pathogenesis of NEC. However, these results are difficult to compare to the findings of the previous study or interpret as the effect of acetate, propionate and butyrate was studied together following fermentation rather than on a separate basis. The authors do report, however, that butyrate levels were <9.91 μg/mL, propionate levels were 36.68 ± 4.83 μg/mL and acetate levels were 1783.82 ± 43.61 μg/mL following fermentation, suggesting that harmful effects might be associated with high acetate (and to a lower extent propionate), while protective effects might be linked to high butyrate levels in NEC [68]. A study by Nafday et al. suggests that higher concentrations of SCFAs applied by colonic instillation in a rat model cause colonic mucosal injury, particularly in the early postnatal period. Acetate, butyrate and propionate or a combination of these SCFAs were instilled at high concentrations of 300 nM, and histologic injury scores in the colon were recorded 24 h later. The severity of mucosal injury decreased as the rats matured, with significant injury on days 3 and 9 but minimal injury by postnatal day 23. However, in comparison to the above studies, the applied concentrations of acetate, butyrate and propionate were beyond those reported by Roy et al. by 10×, >2500× and 600× times, respectively, by far exceeding physiological levels, thus making clinical interpretation challenging [69]. Nevertheless, the above three reports do suggest that SCFAs can reach concentrations toxic to mucosal cells, and various SCFAs have different effects on gut health at various concentrations.

## 8. Future Directions of Research

Key areas of current knowledge gaps and foci of priority in understanding the role of SCFAs in neonatal immune function and health involve the following:The description of how various types and concentrations of SCFAs affect neonatal Treg development and function compared to adults in culture and in vivo.The description of how the direct availability of SCFAs in breastmilk or formula milk, as well as SCFAs produced by HMO breakdown by the components of the microbiome, contribute to SCFA levels and composition in the neonate during the development.The dynamics of transition from HMOs to dietary fibre as fuel for SCFA production and the associated changes in microbiome composition during the weaning period and its implications on Treg function in the infant.The role of SCFAs in addressing inflammatory complications in preterm and term neonates, such as NEC, bronchopulmonary dysplasia or sepsis.The therapeutic potential of enteral or parenteral supplementation of SCFAs in neonates.

## 9. Conclusions

Although the role of SCFAs in neonatal health and immune regulation are currently not well understood, they may potentially serve as a valuable future target for therapeutic and dietary intervention with long-term effects on health outcome in preterm and term neonates. While our understanding of their role in the immune system and the function of various lymphocyte subsets is improving in adults, there is a lack of studies with a specific focus on early life effects. Future studies in animal models and in humans are warranted to better understand the full potential of SCFAs on T cell subsets, particularly on Treg cells in neonates, and the importance of these effects throughout the whole lifespan of the individual.

## Figures and Tables

**Figure 1 nutrients-14-03670-f001:**
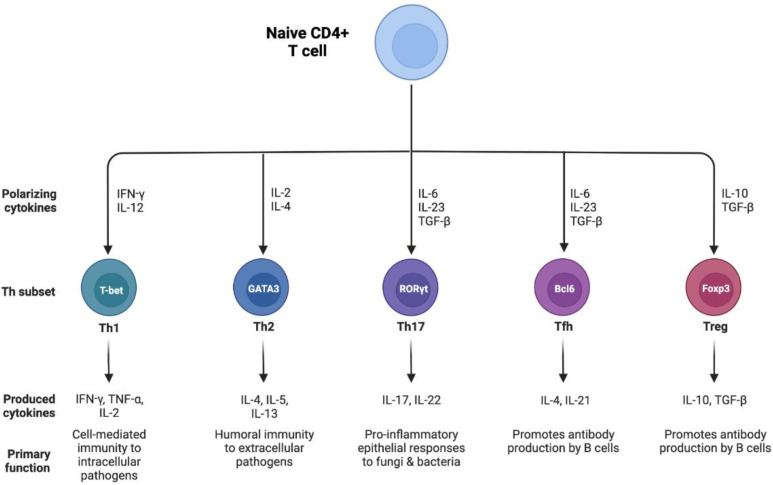
Subsets of CD4+ T helper cells. Th cells differentiate from naive CD4+ T cells into one of five major lineages: Th1, Th2, Th17, T follicular helper (Tfh) and regulatory T (Treg) cells depending on the cytokine milieu in which they are activated. Each Th subset exhibits a unique lineage-defining transcription factor and cytokine expression profile.

## Data Availability

Not applicable.

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
