# Peer review of "The Impact of Short-Chain Fatty Acids on Neonatal Regulatory T Cells"

_nutrients, 2022, doi:10.3390/nu14183670_

Round 1
Reviewer 1 Report
Major point:
The paper is a nice review on the mechanisms how SCA’s can induce the differentiation and expansion of T-regs. For the average reader in the field of nutrition the article is very dense and therefore difficult to follow.
Therefore I would suggest to add to the article a comprehensive infographics or graphical abstract like in their article: Breastfeeding promotes early neonatal regulatory T-cell expansion and immune tolerance of non-inherited maternal antigens, published in Allergy. 2021; 76:2447–2460 in order to make it a review which each reader can enjoy to read it. In the present form the paper is written for experts in adaptive immunology, but not for people interested in the bridge between neonatal nutrition, postnatal development of microbiome and adaptive immunology.
The paper would gain much more attention providing also condensed visual explanations.
Author Response
Thank you for your comments. We have prepared a graphical abstract as well as a figure on T cell subsets as per your suggestion.
Reviewer 2 Report
The review was written well. It can be published without any revision.
Author Response
Thank you for your comments.